# Effects of a High-Fat Diet on Insulin-Related miRNAs in Plasma and Brain Tissue in APP_Swe_/PS1dE9 and Wild-Type C57BL/6J Mice

**DOI:** 10.3390/nu16070955

**Published:** 2024-03-26

**Authors:** Melina Rojas-Criollo, Nil Novau-Ferré, Laia Gutierrez-Tordera, Miren Ettcheto, Jaume Folch, Christopher Papandreou, Laura Panisello, Amanda Cano, Hamza Mostafa, Javier Mateu-Fabregat, Marina Carrasco, Antoni Camins, Mònica Bulló

**Affiliations:** 1Nutrition and Metabolic Health Research Group, Department of Biochemistry and Biotechnology, Rovira i Virgili University (URV), 43201 Reus, Spain; melina.rojas@urv.cat (M.R.-C.); nil.novau@urv.cat (N.N.-F.); laia.gutierrez@iispv.cat (L.G.-T.); jaume.folch@urv.cat (J.F.); christoforos.papandreou@iispv.cat (C.P.); laura.panisello@urv.cat (L.P.); hamza.mostafa@iispv.cat (H.M.); javier.mateu@urv.cat (J.M.-F.); 2Institute of Health Pere Virgili (IISPV), 43204 Reus, Spain; 3Center of Environmental, Food and Toxicological Technology—TecnATox, Rovira i Virgili University, 43201 Reus, Spain; 4Department of Pharmacology, Toxicology and Therapeutic Chemistry, Faculty of Pharmacy and Food Science, Universitat de Barcelona, 08028 Barcelona, Spain; mirenettcheto@ub.edu (M.E.); marinacarrasco@ub.edu (M.C.); camins@ub.edu (A.C.); 5Institute of Neuroscience, Universitat de Barcelona, 08034 Barcelona, Spain; 6Biomedical Research Networking Centre in Neurodegenerative Diseases (CIBERNED), Carlos III Health Institute, 28029 Madrid, Spain; acano@fundacioace.org; 7Ace Alzheimer Center Barcelona, Universitat Internacional de Catalunya, 08028 Barcelona, Spain; 8CIBER Physiology of Obesity and Nutrition (CIBEROBN), Carlos III Health Institute, 28029 Madrid, Spain

**Keywords:** Alzheimer’s disease, high-fat diet, miRNA

## Abstract

Insulin resistance (IR)-related miRNAs have been associated with the development and progression of Alzheimer’s disease (AD). The dietary modulation of these miRNAs could become a potential strategy to manage AD. The aim of this study was to evaluate the effect of a high-fat diet (HFD), which aggravates AD-related pathogenic processes, on serum, cortex and hippocampus IR-related miRNA expression. C57BL/6J WT and APP_Swe_/PS1dE9 mice were fed either an HFD or a conventional diet till 6 months of age. The mice fed with the HFD showed a significant increase in body weight and worsening glucose and insulin metabolism. miR-19a-3p was found to be up-regulated in the cortex, hippocampus and serum of APP/PS1 mice and in the serum and hippocampus of WT mice fed with the HFD. miR-34a-5p and miR-146a-5p were up-regulated in the serum of both groups of mice after consuming the HFD. Serum miR-29c-3p was overexpressed after consuming the HFD, along with hippocampal miR-338-3p and miR-125b-5p, only in WT mice. The HFD modulated the expression of peripheral and brain miRNAs related to glucose and insulin metabolism, suggesting the potential role of these miRNAs not only as therapeutic targets of AD but also as peripheral biomarkers for monitoring AD.

## 1. Introduction

Metabolic diseases such as obesity and type 2 diabetes (T2D) are recognized risk factors for dementia [1]. Dementia is characterized by several symptoms, which include progressive loss of memory and disturbance in normal behavior [2]. Among the different types of dementia, Alzheimer’s disease (AD) and vascular dementia (VaD) stand out as the most prevalent [3]. Obesity entails several detrimental consequences in the brain, including impaired synaptic plasticity, reduced brain volume, low-grade inflammation and neurodegeneration, that can contribute to AD and other types of dementia [4]. Obesity, but especially central obesity, is commonly found in patients with T2D and it is closely associated with insulin resistance (IR) in peripheral tissues [5]. Furthermore, central obesity accounts for structural abnormalities in the brain, cognitive impairment and the development of dementia irrespective of the obesity degree [6]. Excess of body weight is also related to the occurrence of brain IR, and although peripheral and brain insulin resistance are not always directly connected, these two conditions are intertwined and clearly associated with AD [7].

Although the exact mechanisms explaining the role of IR in AD development are largely unknown, animal studies demonstrated that brain insulin improved hippocampal synaptic plasticity and remodeling by increasing GluA1 palmitoylation through FoxO3a [8]. Insulin also exerts beneficial effects on neurons via the AKT and MAPK signaling pathways [9]. Conversely, brain insulin resistance induces Tau hyperphosphorylation and aggregation [10] and the formation of amyloid beta (Aβ) fibrils by inducing the clustering of the GM1 ganglioside in presynaptic membranes [11]. In turn, there is a reciprocal relationship between AD pathology and brain insulin that can lead to positive feedback, with further exacerbation of metabolic dysfunctions. Within these complex interactions, microRNAs (miRNAs) might constitute a molecular link between metabolic conditions and AD. miRNAs are a class of evolutionarily conserved small non-coding RNA molecules with a length of 19–25 nucleotides, which are involved in gene regulation by mediating the degradation of mRNA and affecting its translation to proteins. Beyond their intracellular role, an important proportion of miRNAs migrates outside of the cell and are delivered into bodily fluids, where they are transported either in association with proteins, which is estimated to concern around 90% of the total miRNAs, or in exosomes [12]. The stability of miRNAs in the extracellular environment makes them good biomarkers for various human disorders.

miRNAs, such as miR-29a-3p, miR-34a, miR-146a-5p [13], regulate many biological processes including neurogenesis, dendritic spine morphology and synaptic plasticity and are widely present throughout the nervous system. Others, such as miR-125b-5p and miR-20b-5p, found in high levels in the hippocampus, were associated with memory impairment [14,15]. Several miRNA signatures, discriminating AD dementia patients from individuals without AD dementia or predicting the conversion in patients from mild-cognitive impairment to AD dementia with high accuracy, were described [16,17]. miRNAs, including miR-29a-3p, miR-34a, miR-125b-5p and miR-20b-5p, within others, have been also related to insulin secretion, beta cell development and the regulation of the insulin signaling pathway [18,19,20,21]. Similarly, dysregulated levels of miR-155-5p and miR-98-5p in the serum were also connected with neurological deficits [22,23] and the dysregulation of insulin signaling [24,25,26] and glucose metabolism [27,28]. Decreased plasma levels of some of these specific miRNAs, such as miR-20b-5p, miR-125b-5p, miR-206, miR-29c and miR-98-5p, were also related to neurogenesis, inflammation, cytoskeleton rearrangement, axon guidance, oxidative stress Aβ plaque deposition [15,22,29,30] and up-regulated BACE1 [31,32]. Additionally, some of these miRNAs were shown to play a role in regulating insulin-like growth factor 2 mRNA-binding protein 2 (IGF2BP3) [33] and HbA1c and glucose levels [24], as well as the insulin and MAPK signaling pathways [25,34]. Therefore, insulin resistance-related miRNAs could be the missing link between T2D and Alzheimer’s disease, and strategies focused on modulating these miRNAs would offer a useful approach to ameliorate metabolic disturbances and potentially prevent and treat Alzheimer’s disease [35]. Altered expression of several miRNAs was reported within the hippocampus and cortex in animal models of AD [36]. However, due to the invasive nature and difficulties of accessing brain tissue in living patients, circulating miRNAs constitute a more reliable tool for future studies. Nevertheless, peripheral miRNAs may not fully represent the epigenetic activity in different brain regions, and a co-expression analysis of miRNAs in different tissue types is needed to better understand cross-tissue relationships.

To address these challenges, in this study, we aimed to evaluate the impact of a high-fat diet (HFD), which aggravates the metabolic status and AD-related pathogenic processes, on IR-related miRNA expression in the serum, cortex, and hippocampus of APP/PS1 mice compared to wild-type (WT) mice. We also explored whether changes in brain tissue miRNA expression, either in the cortex or in the hippocampus, corresponded to changes in miRNA levels in peripheral blood.

## 2. Materials and Methods

### 2.1. Animals and Dietary Treatment

APP_Swe_/PS1dE9 (APP/PS1) double-transgenic male mice that express a Swedish double mutation (K594M/N595L) of a chimeric mouse/human APP (mo/huAPP695_swe_) and the exon-9-deleted PSEN1 gene (PSEN1-dE9) [37] and C57BL/6 wild-type (WT) littermates were used. All animals were obtained from established breeding couples in the institutional animal facilities of the Faculty of Pharmacy and Food Sciences of the University of Barcelona (approval number C-0032). After weaning, at 21 days of age, both WT and APP/PS1 animals were randomly allocated to a control group or a high-fat-diet group and fed the corresponding diets for 6 months. In the control group, the animals were fed with conventional chow (control diet, CD; ENVIGO, Madison, WI, USA), while, in the intervention group, the animals were exposed to a 60% HFD predominantly sourced from hydrogenated coconut oil and rich in palmitic acid (Research Diets Inc., New Brunswick, NJ, USA). The animals were housed under controlled room temperature and humidity on a 12:12 h light–dark cycle with access to water and food ad libitum. They were regularly weighted, and food consumption was monitored. The mice were sacrificed by cervical dislocation. Immediately after this, samples of serum, cortex and hippocampus were obtained, frozen and stored at −80 °C until further processing. The miRNA analyses were conducted in blinded conditions. Every possible effort was made to reduce the number of animals used and to minimize their suffering. All experiments were approved by the institutional ethical committee on 30 June 2021. The mice were treated in accordance with the European Community Council Directive 86/609/EEC and the procedures established by the Departament d’Agricultura, Ramaderia i Pesca of the Generalitat de Catalunya.

### 2.2. Glucose and Insulin Tolerance Tests

The intraperitoneal glucose tolerance test (GTT) and insulin tolerance test (ITT) were performed as previously described [38] in 8 animals from each group. The mice were fasted at least for 6 h prior to carrying out both tests. For the GTT, blood samples were obtained from the tail 30 min prior to a 1 g/kg intraperitoneal glucose injection. The ITT was performed in similar conditions, using 0.25 IU/kg of human insulin diluted in saline (Humulina Regular, 100 IU/mL/Lilly, S.A.; Madrid, Spain). The glucose levels were measured (Accu-chek^®^ Aviva glucometer, Roche, Mannheim, Germany) 0, 5, 15, 30, 60, 120 and 180 min after glucose administration and 0, 15, 30, 45, 60 and 90 min after insulin administration. If the blood glucose levels dropped below 20 mg/dL during the procedure, 1 g of glucose/kg was additionally administered.

### 2.3. RNA Extraction and cDNA Synthesis

Total RNA was extracted from the serum, cortex and hippocampus samples from at least 11 animals per each experimental group using the mirVANA PARIS kit, following the manufacturer’s protocol (Ambion^®^-Life Technologies, Carlsbad, CA, USA). Before the extraction, the cortex and the hippocampus samples were homogenized (IKAᵀᴹ ULTRA-TURRAXᵀᴹ T 18 Digital Disperser, IKA Werke GmbH & Co., Staufen, Germany) using the Cell Disruption Buffer provided in the mirVANA PARIS kit. RNA quantity and quality (260/280, 260/230) were measured by a NanoDrop 2000 spectrophotometer (Thermo Fisher Scientific, Waltham, MA, USA) in 2 µL of each sample to ensure the integrity and purity of RNA prior to further analyses. Total RNA was reverse-transcribed to cDNA using the TaqMan MicroRNA Reverse Transcription kit in a GeneAmp PCR System 9700 thermocycler (Applied Biosystems, Darmstadt, Germany), following the manufacturer’s instructions. Real-time PCR was performed in a 7900HT Fast Real-Time PCR System (Applied Biosystems, Darmstadt, Germany) using cel-miR-39-3p from *Caenorhabditis elegans* as an exogenous control and ath-miR-159a from *Arabidopsis thaliana* as a negative control (Life Technologies Corporation, Pleasanton, CA, US). miR-16-5p and miR-30e-5p were initially considered as housekeeping miRNAs. The primers used in this analysis were designed specifically for TaqMan Gene Expression Assays (Thermo Fisher Scientific). All measurements were performed in duplicate, and the qPCR data were processed using 7900 SDS v2.4.1 software.

The selection of the analyzed miRNAs was based on a comprehensive literature review conducted in 2022, considering those miRNAs expressed in mice and preferably also in humans that were related to AD or to hallmarks of AD (β-amyloid, tau, p-tau, neurodegeneration) together with insulin resistance or T2D. After completing further refinement stages, a total of 15 miRNAs were considered for validation in all samples.

### 2.4. Functional Enrichment Analysis

The minimum network of the target genes of miRNAs with the strongest evidence of being either linked to neurodegeneration in APP/PS1 mice or affected by the exposure to the high-fat diet was identified through the validated miRTarBase [39] and TarBase [40] databases, incorporated in the miRNet platform [40,41]. Subsequent Gene Ontology (GO) analysis, which covered biological process (BP), molecular function (MF) and cellular component (CC), as well as Kyoto Encyclopedia of Genes and Genomes (KEGG) pathway analysis, were conducted in the WebGestalt platform to elucidate the potential functions and pathway enrichment associations (significant as FDR < 0.05) [42]. For enhanced reliability, genes from the verified miRTarBase database, in conjunction with genes that overlapped in the predicted miRDB and TargetScan databases were employed.

### 2.5. Statistical Analysis

The sample size was estimated to detect a difference of at least 1.5-fold, with >90% of statistical power (α = 0.05; two-sided). This fixed fold change cut-off was established based on a review of the existing literature, to identify the genes exhibiting the most significant variation [43,44]. Quantitative variables in the descriptive analysis are expressed as mean ± standard deviation (SD). Normal distribution was assessed using the Shapiro–Wilk test. Group differences were examined using the Student’s *t* test and the Mann–Whitney test for normally and not normally distributed data, respectively. Differences were considered significant at *p* < 0.05.

Since miR-16-5p and miR-30e-5p did not display stability according to the dietary treatment, the cycle threshold (Ct) values of each miRNA were mean-centered using the exogenous oligonucleotide cel-miR-39, and the 2 miRNAs, although not following the initial selection criteria, were finally considered as part of the results. Lack of miRNA expression was considered when Ct > 35. When the Ct values were higher than 35 in more than 80% of the samples, the corresponding miRNA was not considered for the subsequent analyses.

Then, a second normalization was performed by mean-centering the values using their respective means in control mice. Finally, for the expression data of the remaining miRNAs, we calculated the log2 fold change (log2FC) for the downstream analyses. In all samples, we added a negative control miRNA (ath-miR-159), and if this negative control was expressed, the sample was removed from the analyses. Extreme outliers were detected and excluded if they were outside the intervals Q1 − 3 × IQR and Q3 + 3 × IQR, with Q1, Q3 and IQR being the first and third quartiles and the interquartile range, respectively.

Differentially regulated miRNAs were identified by the Student’s *t* test or the Mann–Whitney test depending on the data distribution. *p*-values were considered significant when *p* < 0.05 after Benjamini–Hochberg correction. For the statistical tests, normalized relative log2 ratios were employed. The antilog-transformed values are reported as fold changes between groups for each comparison. miRNA analyses were performed, and all graphs were obtained using R version 4.2.1 (R Foundation for Statistical Computing, Vienna, Austria).

## 3. Results

As expected, the mice fed with the HFD showed an increased body weight when compared to the mice fed with conventional chow (*p* = 0.0024 for WT, *p* ≤ 0.0001 for APP/PS1) (Figure 1A). Similarly, following the HFD induced alterations in peripheral glucose metabolism, as evidenced by both GTT (Figure 1B,C) and ITT (Figure 1D,E). Results showing the deleterious effect of following an HFD on AD pathology according to the genotype were previously published by our group [45,46,47]. Briefly, feeding APP/PS1 mice with an HFD aggravated the animals’ learning and memory abilities together with increasing neuroinflammation, brain amyloid β (Aβ) production and plaque burden.

No expression of miR155-5p was observed either in the cortex or in the hippocampus. miR98-5P was not expressed in the serum, whereas miR206-3p and miR20b-5p were not found expressed in the cortex. miR20b-5p, miR130b-3p and miR 206-3p were not expressed in the hippocampus samples.

### 3.1. Differential Expression of miRNAs in the Serum, Cortex and Hippocampus of APP/PS1 vs. WT Mice

In Figure 2, we show the differential expression of miRNAs in the serum, cortex and hippocampus between wild-type and APP/PS1 mice. In the serum, miR-19a-3p was found to be up-regulated, and miR-34a-5p and miR-155-5p were down-regulated in APP/PS1 mice compared to WT animals.

However, these differences were not significant after accounting for multiple comparisons (Appendix A). In the cortex, miRNA-19a-3p was also up-regulated, and miR-30-5p was down-regulated in APP/PS1, but only miR-19a-3p remained significantly up-regulated after adjusting by false discovery rate (Figure 2B, Appendix A). Several miRNAs were differentially expressed in the hippocampus according to the mice genotype. After adjustment for multiple testing, miR-9-5p, miR-16-5p, miR-19a-3p, miR-22-3p, miR-29c-3p, miR-181c-5p and miR-338-3p were found to be up-regulated in APP/PS1 mice (Figure 2C, Appendix A). miR-19a-3p was consistently up-regulated in APP/PS1 mice, irrespective of the tissue. miR-98-5p was not found expressed in the serum and was detected only in the cortex and hippocampus, and miR-130b-3p was not expressed in the hippocampus. miR-20e-5p, miR-155-5p and miR-206-3p were not expressed either in the cortex or in the hippocampus.

### 3.2. Effect of the HFD on Serum, Cortex and Hippocampus miRNA Expression

The changes in miRNA expression in the serum, cortex and hippocampus in animals fed with the high-fat diet compared to those receiving regular chow are shown in Figure 3. In the serum, miR-19a-3p, miR-20b-5p, miR-22-3p, miR-29c-3p, miR-30e-5p, miR-34a-5p, miR-130b-3p and miR-146a-5p were up-regulated in WT mice after consuming the high-fat diet (Figure 3a, Appendix A), whereas only miR-34a-5p and miR-146a-5p were up-regulated in APP/PS1 animals after the dietary intervention (Figure 3d, Appendix A). The high-fat diet did not induce significant changes in miRNA expression in the cortex (Figure 3b,e; Appendix A). Only WT mice fed with a high-fat diet displayed a significant up-regulation of miR-19a-3p, miR-29c-3p, miR-125b-5p and miR-338-3p in the hippocampus (Figure 3c, Appendix A).

### 3.3. Functional Enrichment Analysis of miRNAs Differentially Modulated by the HFD

We performed functional enrichment analyses for miRNAs differentially expressed according to the genotypes or up- and down-regulated either in the serum or in brain tissues in APP/PS1 or WT animals. The gene–miRNA interaction analysis identified 33 hub genes (Figure 4A). The identified miRNAs regulated genes involved in the MAPK signaling pathway, axon guidance, neurogenesis, programmed cell death and the PI3K-Akt signaling pathway (Figure 4B and Appendix A). Some of the molecular functions involved ubiquitin-like protein transferase activity and cytoskeletal protein binding and the cellular component of neurons (Appendix A). 

## 4. Discussion

In the present study, we observed the consistent up-regulation of miR-19a-3p in the serum, cortex and hippocampus of APP/PS1 mice and in the serum and hippocampus of wild-type mice fed with a high-fat diet. Furthermore, we found that the high-fat diet up-regulated the serum and hippocampal miR-29c-3p, along with the hippocampal miR-338-3p and miR-125b-5p in WT mice, whereas miR-34a-5p and miR-146a-5p were up-regulated in the serum of both genotypes after HFD consumption. These findings support an active role of these miRNAs in the progression of neurodegeneration aggravated by the metabolic misbalance produced by consuming a high-fat diet.

Several animal studies demonstrated that HFDs not only enhance and accelerate cognitive impairment symptoms and Alzheimer’s disease hallmarks in AD rodent models, but also serve as a potential AD-initiating factor by inducing neurodegeneration hallmarks in wild-type animals [48,49]. Although the mechanisms that precede and trigger brain damage during the silent pre-clinical period are not well understood, HFD-induced systemic metabolic alterations, including dysregulation of glucose and insulin metabolism, have emerged as promising factors driving pre-clinical AD. As previously described [50], in our study the animals fed with the high-fat diet showed increased weight and worsened peripheral glucose and insulin metabolism. This metabolic worsening was accompanied by the differential expression of several miRNAs in the circulation and in brain tissues.

A previous study found that miR-19a-3p was down-regulated in diabetic patients, with its plasma levels negatively correlated with the blood glucose levels [26]. In vitro experiments in pancreatic β-cells demonstrated that the overexpression of miR-19a-3p enhanced cell proliferation and insulin secretion and inhibited apoptosis, supporting miR-19a-3p as a candidate for managing type 2 diabetes [26]. Furthermore, miR-19a-3p was found to be down-regulated in the cerebrospinal fluid in Parkinson’s disease (PD) patients, and its levels appeared inversely related to cerebrospinal fluid (CSF) Aβ plaque density in both AD and PD patients [51]. Similarly, the serum levels of miR-19b-3p, which belongs to the same miR-17/92 cluster as miR-19a-3p, were found to be significantly lower in AD patients compared to control subjects [52]. Notably, its overexpression was shown to alleviate Aβ-induced injury by targeting β-secretase (BACE1) [53], an aspartyl protease of the pepsin family whose concentrations and rates of activity are increased in AD brains and body fluids. However, there is a controversy regarding miR-19b-3p, as its up-regulation was found to exacerbate abnormal synaptic plasticity and cognitive impairment in mice, while its inhibition rescues synaptic transmission and plasticity in hippocampal neurons, improving abnormal dendritic structures [54]. Although there is no information regarding functional similarities between these two miRNAs, our results support a deleterious role of miR-19a-3p in the development of neurodegeneration. The up-regulation of miR-19a-3p observed in our study in mice after consuming the HFD, concomitantly with increased body weight and worsened glucose and insulin metabolism, might indicate a mechanism linking peripheral metabolic dysregulation to central metabolic and neurodegenerative worsening. This is further supported by the functional enrichment analysis in which miR-19a-3p targets mitogen-activated protein kinase (MAPK) signaling pathways, neuron differentiation and protein phosphorylation pathways. Moreover, we observed a significant up-regulation of miR-29c-3p in the hippocampus of APP/PS1 mice and in both the hippocampus and the serum of WT mice fed with the HFD. The miRNA-29 family is known to target BACE1, exerting a negative regulation on Aβ formation [55,56,57]. Specifically, miR-29c-3p binds the 3′-untranslated region (3′-UTR) of BACE1, down-regulating its expression and affecting the progression to AD. This miRNA was reported to be abnormally expressed in various diseases and down-regulated in AD [58]. Similarly, the miR-29a/b-1 cluster was shown to be down-regulated, corresponding to increased BACE1 levels in post-mortem brain samples of idiopathic AD patients [57]. However, the expression of the microRNA-29 family is consistently increased in different tissues in several metabolic conditions including obesity, insulin resistance and type 2 diabetes. Therefore, the overexpression we found of miR-29c-3p in mice fed with the high-fat diet, which also displayed an altered metabolic status (obesity and glucose derangements), could be interpreted as a counterregulatory mechanism against the deleterious effects of such a diet on brain.

In the present study, we found miR-146a-5p and miR-34a up-regulation in the serum of both WT and APP/PS1 mice fed with the high-fat diet. miR-146a-5p has been implicated in neuroinflammation processes, including Aβ deposition and synaptic pathological changes [59]. This miRNA may also play a modulatory role in insulin secretion, glucose homeostasis, adipocyte differentiation and cell proliferation [60,61], indicating a mechanistic link between insulin metabolism and neuroinflammatory diseases. Clinical studies reported a higher expression of miR-146a-5p in the brain tissue of AD patients [22]. Preclinical and clinical trials also highlighted the role of this miRNA in the pathogenesis of AD, suggesting its utility as a potential therapeutic target for AD. A recent study conducted on APP/PS1 mice, reported increased levels of miR-146a-5p in the hippocampus, and the administration of its antagomir rescued neurogenesis and pattern separation [62]. In our study, miR-146a-5p showed a non-significant trend toward up-regulation in the hippocampus of APP/PS1 mice. However, this miRNA was significantly up-regulated in the serum of both WT and APP/PS1 mice after feeding with the high-fat diet. We also found a significant up-regulation of miR-146a-5p in the cortex of APP/PS1 fed with the HFD, although it was attenuated after adjusting for multiple comparisons. Similarly, miR-34a was also found to be up-regulated in the serum of WT and APP/PS1 mice after consuming the HFD. Increased miR-34a expression is associated with cognitive impairment and AD-like pathology by targeting alpha secretase (ADAM10), NMDAR2B and SIRT1 RNAs, whose levels are significantly reduced by miR-34a overexpression [63]. Interestingly, the mature miR-34a is one of the major miRNAs involved in insulin production and glucose homeostasis [64]. Clinical studies reported higher circulating levels of this miRNA in pre-diabetic and diabetic individuals compared to normoglycemic subjects [65]. Therefore, its up-regulation in the serum of the examined mice after consuming the HFD supports the role of this miRNA in the development of AD-related features and its potential as a nutritional therapeutic target.

This study has some strengths and limitations. The study evaluated miRNA expression at the peripheral and brain tissue levels in the same animals, thus providing a comprehensive knowledge of the cross-tissue relationships and potentially identifying peripheral biomarkers or brain epigenetic activity. Whereas brain tissue miRNA expression would indicate epigenetic changes at the brain level, the miRNA circulating levels could be interpreted as biomarkers of brain damage. However, the nature of this study does not allow to discard the possibility that circulating miRNAs could effectively cross the blood–brain barrier and regulate brain damage. As regards the limitations, this study primarily examined plasma and brain tissues without investigating other relevant tissues such as adipose tissue and liver in relation to insulin resistance. Since it was conducted only in males, we cannot discard a potential sex-dependent differential regulation by HFD of miRNA expression, and our results should be confirmed in female mice. Finally, due to the lack of validation experimental methods to evaluate changes in pathways and target gene expression, our results have to be interpreted with caution.

## 5. Conclusions

In summary, our findings support the role of specific insulin-related miRNAs in the development of Alzheimer’s specific features and indicate the potential of a high-fat diet to aggravate neurodegenerative processes. Notably, miR-19a-3p and miR-29c-3p showed similar variations in both peripheral and central levels after high-fat diet consumption, suggesting their potential role not only as therapeutic targets for Alzheimer’s disease but also as peripheral biomarkers of the disease.

## Figures and Tables

**Figure 1 nutrients-16-00955-f001:**
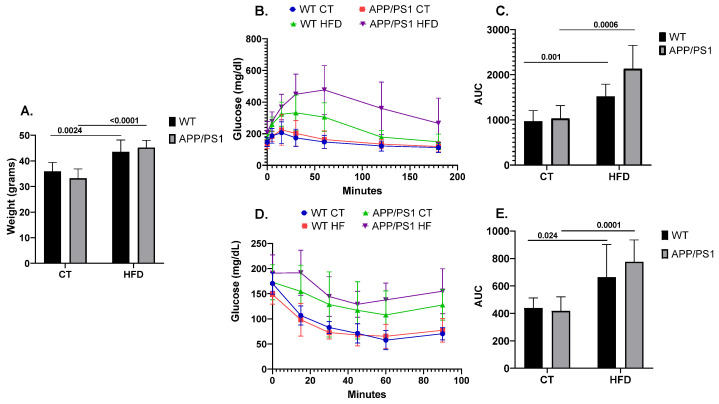
Weight, glucose and insulin metabolism in mice (*n* = 8 animals per group). (**A**) Comparison of body weight among the different experimental groups, (**B**) Glucose Tolerance Test (GTT) and (**D**) Insulin Tolerance Test’s (ITT) experimental profiles. Area under the curve (AUC) was calculated from the time point 0 until the end of the experiment for both (**C**) GTT and (**E**) ITT tests. Student’s *t* test or Mann–Whitney test was used. All results are presented as mean ± SD.

**Figure 2 nutrients-16-00955-f002:**
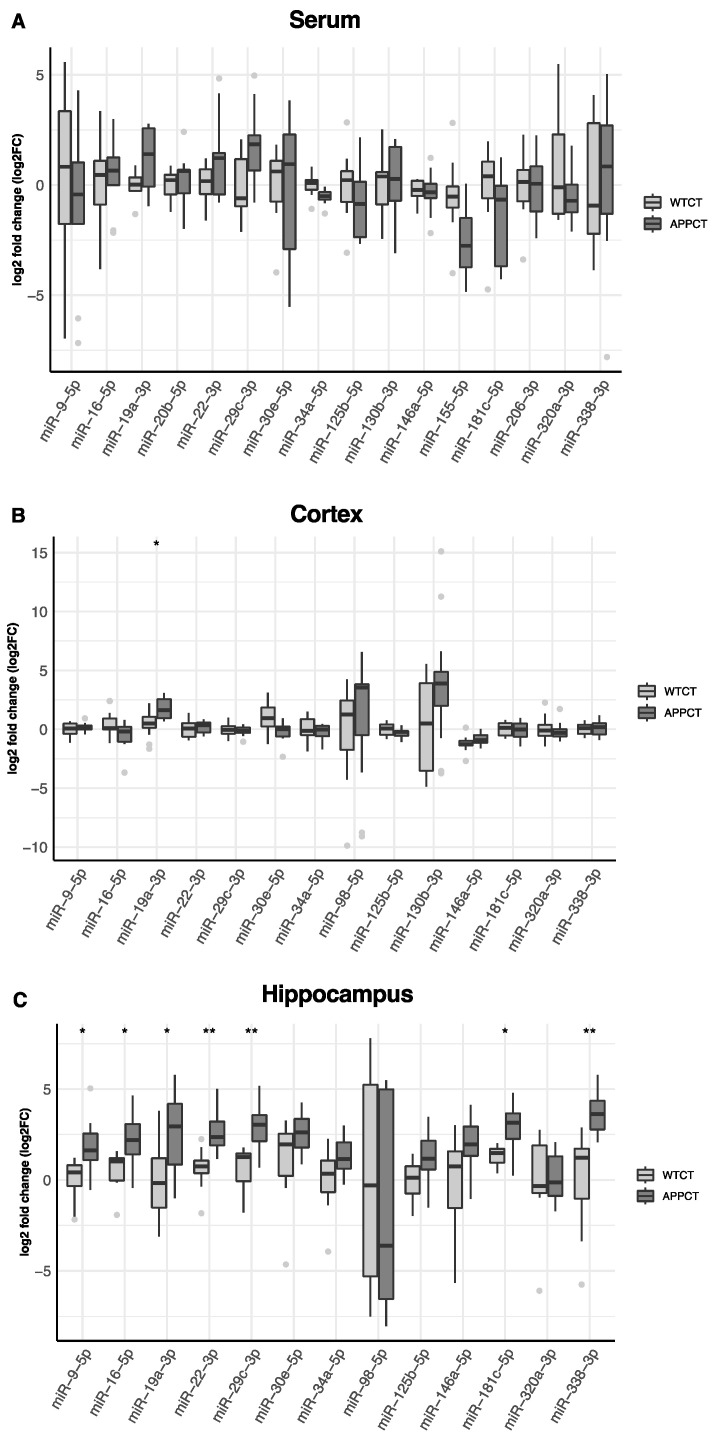
Differences in the expression of miRNAs in WT and APP/PS1 mice. (**A**) serum, (**B**) cortex and (**C**) hippocampus. At least 11 animals were considered in each experimental group. Student’s *t* test or Mann–Whitney test and Benjamini–Hochberg correction (FDR, * Padj < 0.05, ** Padj < 0.01) were performed. Grey light dots represent outliers.

**Figure 3 nutrients-16-00955-f003:**
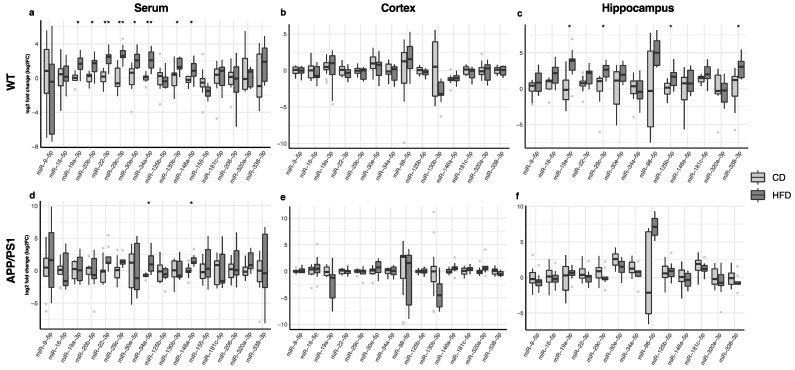
Differences in the expression of miRNAs in WT and APP/PS1 mice fed with the high-fat diet compared to their counterparts fed with regular chow. (**a**,**d**) Serum, (**b**,**e**) cortex and (**c**,**f**) hippocampus. At least 11 animals were considered in each experimental group. Student’s *t* test or Mann–Whitney test and Benjamini–Hochberg correction (FDR, * Padj < 0.05, ** Padj < 0.01) were performed. Grey light dots represent outliers.

**Figure 4 nutrients-16-00955-f004:**
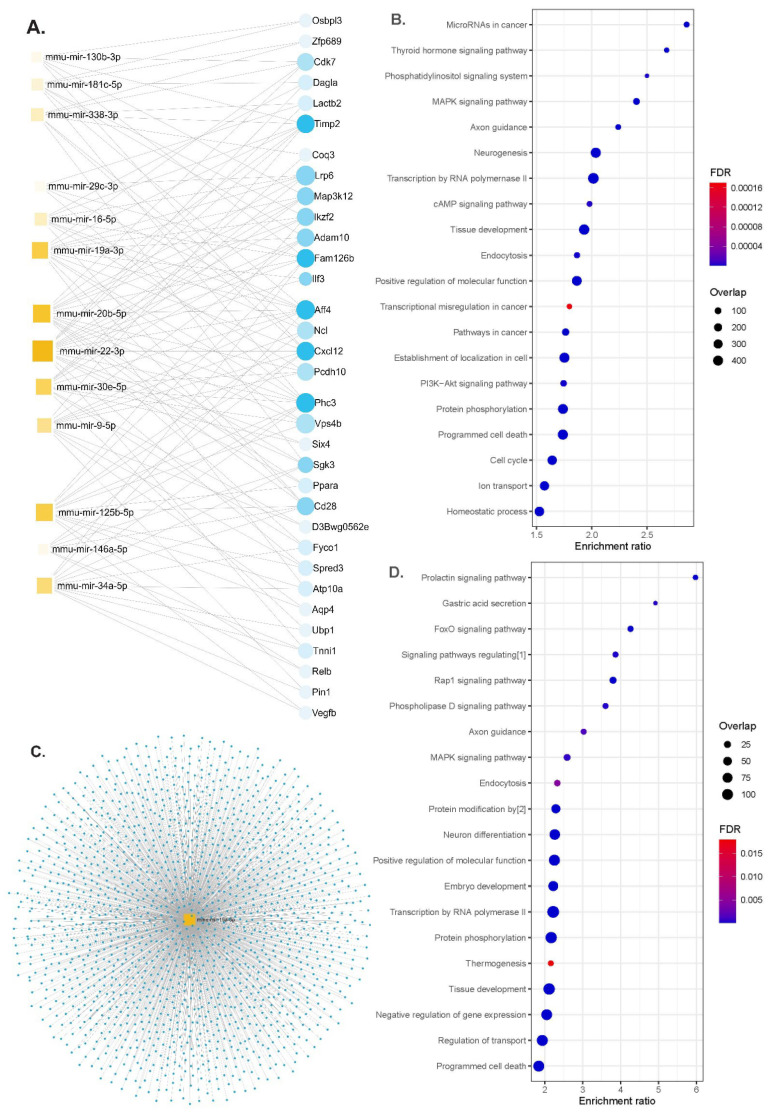
Functional enrichment analyses. (**A**) Network diagram of 13 miRNAs and potential target genes (blue dots are nodes representing the differentially expressed genes, and squares indicate the miRNAs; their size indicates the number of targeted genes), (**B**) significant KEGG pathway and GO enrichment analysis of candidate hub genes on the predicted targets of each up-regulated miRNA (dot size reflects the number of genes in each GO pathway, FDR, Padj < 0.05), (**C**) the network diagram of mmu-miR-19a-3p overlaps for serum and brain tissues (blue dots for target genes), (**D**) significant KEGG pathways and GO functions for the target genes of mmu-miR-19a-3p (dot size reflects the number of genes in each GO pathway, FDR, Padj < 0.05). Abbreviations: [1] signaling pathways regulating pluripotency of stem cells; [2] protein modification by small protein conjugation or removal, FDR; false discovery rate.

## Data Availability

The data presented in this study are available on request from the corresponding author due to privacy.

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
