# Peer review of "Effects of a High-Fat Diet on Insulin-Related miRNAs in Plasma and Brain Tissue in APPSwe/PS1dE9 and Wild-Type C57BL/6J Mice"

_nutrients, 2024, doi:10.3390/nu16070955_

Round 1

Reviewer 1 Report

Comments and Suggestions for Authors

In order to investigate the effects of a high-fat diet, the study compared insulin-related miRNAs in the brain tissue and plasma of wild-type (WT) and APP/PS1 mice, a model for Alzheimer's disease (AD). They discovered that the two groups' serum, cortical, and hippocampal regions expressed miRNAs differently. In APP/PS1 mice relative to WT, there was an up-regulation of miR-19a-3p in serum, but a down-regulation of miR-34a-5p and miR-155-5p. Tests for insulin and glucose tolerance revealed that the mice fed a high-fat diet had altered glucose metabolism and significantly gained weight. Furthermore, in the context of AD, their data imply a potential connection between diet-induced metabolic alterations, miRNA expression, and insulin regulation. But in my opinion, despite the fact that all of the studies purport to be on AD, the lack of pathological alterations specific to AD significantly diminishes the significance of this research. Furthermore, it's important to understand why miRNAs choose their targets. It is advised that more pertinent data be gathered and that a more thorough mechanistic theory be put forth to explain how HFD alters the course of AD by influencing these miRNAs. Below are some remarks that are attached. 

1.     Figure 1: At what age were the mice in this experiment, and what was the duration of their HFD feeding? Furthermore, it is difficult to tell the groups in Figs. 1B and 1C apart. It is advised that they be replaced with colored symbols.

2.     Figure 2: It appears that the authors only conducted quantitative analysis on the 13 (or 17) particular miRNAs listed in the figure, despite the fact that there are thousands of known types of miRNAs. As a result, I believe the authors need to give adequate justification and supporting data for their choice of miRNAs. In fact, I believe that the authors' selection process might be more convincingly justified if they could include pertinent RNAseq results as supplemental data.

3.     Fig. 3: The authors deduced changes in each miRNA and we can understand the effect of HFD on different tissues based on these results. Although the authors reported changes in different tissues, we can see that different tissues exhibit different changes even when the same miRNA is used. For instance, there are variations in miR-34a-5p in the hippocampal, cortex, and serum. And how does this affect how AD develops? A plausible mechanism explanation from the authors is required.

4.     Fig. 4: I'm wondering if the associated target genes have changed as predicted in Table S1 based on a comparison with Supplementary Table S1. It would be preferable if the results could be shown in this figure.

5.     After being fed a high-fat diet, do these mice—especially the AD mice—have any changes in their cognitive function? Furthermore, it appears that the authors did not examine pathological variations in tissues, particularly the brain. I believe that the absence of narrative in this field will continue to undermine its ability to draw conclusions about AD.

Comments on the Quality of English Language

Only minor editing of English language is required.

Reviewer 2 Report

Comments and Suggestions for Authors

1. Introduction: The introduction could provide more background on the specific miRNAs studied and their known roles in insulin resistance and AD. Could you provide more information on the selection criteria for the miRNAs studied and their relevance to insulin resistance and AD? What is the rationale behind selecting these specific miRNAs for the study?

2. Were both male and female mice considered for this study, and if not, what was the rationale for using only male mice?

3. The manuscript mentions that 17 miRNAs were considered for analysis based on a literature review. However, it is not clear how these miRNAs were specifically linked to AD or insulin resistance. Can you elaborate on the selection process for the 17 miRNAs and their direct relevance to AD and insulin resistance?

4. The results show that mice fed with a high-fat diet had increased body weight and altered glucose and insulin metabolism. However, the manuscript could benefit from a more detailed discussion of how these changes correlate with the observed miRNA expression changes.

  • Can you discuss how the changes in body weight and glucose/insulin metabolism might be related to the observed changes in miRNA expression?

5. The discussion provides a good overview of the findings and their implications. However, it would be beneficial to include more information on the limitations of the study and potential future directions.

  • Question: What are the limitations of this study, and how could future research build on these findings to further elucidate the role of miRNAs in AD and insulin resistance?

6. What is the housekeeping gene used for qPCR of miRNA expression? Did researchers used same housekeeping genes for mouse brain tissue and serum? Please explain properly. 

7.  The researchers have depicted the relationships between 13 miRNAs and their potential gene targets. While the diagram illustrates the interactions, there is a lack of detailed information on the validation of these connections through techniques such as Western blotting for protein expression or PCR for gene expression. The assumption that these genes are involved in Alzheimer's disease is primarily based on previous studies, without direct experimental evidence provided in this work. Could you please provide the protein expression or gene expression changes in miRNA targeted genes?

Round 2

Reviewer 1 Report

Comments and Suggestions for Authors

The authors have responded appropriately to all questions I raised.

Author Response

We thank the Reviewer for the helpful comments during the review process.

Reviewer 2 Report

Comments and Suggestions for Authors

The authors have satisfactorily addressed my comments and suggestions. However, I believe the introduction section, specifically the rationale for selecting the miRNAs for the study, requires further strengthening. More effort is needed in this regard.

Author Response

We would like to thank the Reviewer for the comments that contribute to improve the quality of our manuscript. According to these comments, in the introduction section. we have added additional information related to the selected miRNAs. 
